# Malignant Transformation of Giant Cell Tumour of Bone: A Review of Literature and the Experience of a Referral Centre

**DOI:** 10.3390/ijms231810721

**Published:** 2022-09-14

**Authors:** Sabrina Vari, Federica Riva, Concetta Elisa Onesti, Antonella Cosimati, Davide Renna, Roberto Biagini, Jacopo Baldi, Carmine Zoccali, Vincenzo Anelli, Alessio Annovazzi, Renato Covello, Andrea Ascione, Beatrice Casini, Virginia Ferraresi

**Affiliations:** 1UOSD Sarcomas and Rare Tumors, IRCCS Regina Elena National Cancer Institute, 00144 Rome, Italy; 2UOSD Sarcomas and Rare Tumors, IRCCS Regina Elena National Cancer Institute, Sapienza University of Rome, 00144 Rome, Italy; 3Oncological Orthopaedics Unit, IRCCS Regina Elena National Cancer Institute, 00144 Rome, Italy; 4Radiology Department, IRCCS Regina Elena National Cancer Institute, 00144 Rome, Italy; 5Nuclear Medicine Department, IRCCS Regina Elena National Cancer Institute, 00144 Rome, Italy; 6Department of Pathology, IRCCS Regina Elena National Cancer Institute, 00144 Rome, Italy; 7Department of Radiological, Oncological and Pathological Sciences, Sapienza University of Rome, Policlinico Umberto I, 00161 Rome, Italy

**Keywords:** giant cell tumour of bone, bone sarcoma, denosumab, radiological features, diagnosis, malignant transformation, H3F3A

## Abstract

Giant cell tumour of bone (GCTB) is a benign, locally aggressive primary bone neoplasm that represents 5% of all bone tumours. The principal treatment approach is surgery. Although generally GCTB is considered only a locally aggressive disease, it can metastasise, and lung metastases occur in 1–9% of patients. To date, only the use of denosumab has been approved as medical treatment for GCTB. Even more rarely, GCTB undergoes sarcomatous transformation into a malignant tumour (4% of all GCTB), but history of this malignant transformation is unclear and unpredictable. Considering the rarity of the event, the data in the literature are few. In this review, we summarise published data of GCTB malignant transformation and we analyse three cases of malignant transformation of GCTB, evaluating histopathology, genetics, and radiological aspects. Despite the rarity of this event, we conclude that a strict follow up is recommended to detect early malignant transformation.

## 1. Introduction

Giant cell tumour of bone (GCTB) is a rare, benign, and locally aggressive primary bone neoplasm that accounts for up to 5% of all bone tumours [1]. Rarely, GCTB reveals malignant behaviour. This circumstance, that is very uncommon in the primary tumour, is usually described following oncological treatment, typically radiotherapy [2]. Malignant histopathological characteristics of GCTB are more similar to a high-grade sarcoma such as undifferentiated sarcoma or osteosarcoma [2]. To evaluate the incidence of GCTB malignant transformation and the impact of treatment on this event, accurate epidemiologic data are needed. This review summarises published data of GCTB malignant transformation (Table 1), with some consideration to the challenges associated with initial diagnosis, clinical behaviour, treatment, and oncologic outcomes observed in patients with GCTB [3,4,5,6,7,8,9,10,11,12]. Moreover, we reported our experience with three cases of progression and malignant transformation of GCTB.

### 1.1. GCTB: Natural History and Therapeutic Approaches

GCTB typically affects the meta-epiphysis of long bones, most commonly the distal femur and proximal tibia, but it can arise anywhere throughout the skeleton. The peak of incidence is between 20 and 40 years of age and no gender-based predilection has been observed [13]. The principal treatment approach of GCTB is surgery, which generally includes extensive curettage or en-bloc resection. Ideally, extensive curettage combined with high-speed burring and local adjuvant treatment should be the first choice due to the possibility of saving the joint adjacent to the tumour, although it may be associated with a relatively high local recurrence rate [14]. En-bloc resection has been associated with a lower risk of local recurrence, but can lead to severe functional impairment and is usually proposed for those tumours presenting extensive bone and soft tissue destruction. In general, the local recurrence rate of GCTB is quite high for curettage alone (27–65% of the cases), decreases for curettage with adjuvant therapy (12–27% of cases), and is very low for en-bloc resection (0–12% of cases) [15,16]. The role of radiotherapy in GCTB is debated and its use was more frequent in the pre-denosumab era, when it was considered a reasonable choice in the case of lesions that could not be fully excised or in specific anatomic sites, such as the sacrum, where surgery was associated with high morbidity. However, radiation has fallen out of favour, due to the possibility of development of radiation-induced sarcomas or secondary malignant transformation of GCTB, and should now be carefully evaluated only in the absence of more active and safe therapeutic approaches [17,18,19,20]. Although generally GCTB is considered only a locally aggressive disease, it can, rarely, metastasise despite maintaining conventional histological features [2,21,22,23]. Lung metastases occur in 1–9% of patients with GCTB, and even though these pulmonary localisations are usually histologically conventional, malignant transformation can occur in less than 1% of cases [21,24]. According to a systematic review involving 242 patients with lung metastases from GCTB, spontaneous regression was observed in 4.5% of patients [25]. Another study reported that 45% (10 out of 22) of patients with lung metastases who were initially managed with the wait-and-see approach, maintained a stable disease [26]. As such, it can be suggested to adopt a wait-and-see approach for lung metastases in selected cases prior to deciding on a specific oncological treatment; however, nodules measuring ≥ 5 mm have a high risk of growth, and caution is required in presence of such nodules [26,27]. To date, only one medical treatment has been approved for GCTB, after the discovery of the crucial role of RANK/RANKL pathway in the pathogenesis of GCTB. Denosumab is a fully humanised monoclonal antibody that specifically inhibits the receptor activator of nuclear factor kappa-B ligand (RANKL), thereby inhibiting osteoclastogenesis and osteoclast-mediated bone destruction [3,28]. Based on the results in safety and efficacy of denosumab in more than 280 patients with complicated GCTB, reported by Chawla S. at al. in a phase II trial, in 2013 the United States Food and Drug Administration (FDA) approved denosumab for the treatment of unresectable GCTB or for patients in whom surgery would result in severe morbidity [7,29].

### 1.2. GCBT and Malignant Transformation: Histopathology and Genetics Aspects

At least 95% of GCTBs are genetically characterised by driver mutations in the H3-3A (H3F3A) gene, mostly H3.3 p.Gly34Trp [30]. The mechanism by which this genetic mutation, affecting the histone tail of a H3.3 Histone A variant, leads to GCTB development is currently undefined. On microscopic examination, GCTB presents a population of neoplastic, relatively monomorphic, mononuclear cells, admixed with a variety of non-neoplastic cells, among which are the characteristic osteoclast-like, multinucleated, giant cells. The giant cells are characterised by very large numbers of nuclei (sometimes even more than 50) and by nuclear morphology analogous to that of neoplastic cells. Extensive necrosis, haemorrhage, clusters of foamy macrophages, and deposition of reactive bone can all be found in the context of a GCTB, especially in the setting of a pathological fracture. As a result, GCTB appears as a polymorphous histological entity with a substantial list of possible differential diagnoses that include giant cell-rich osteosarcoma and aneurysmal bone cyst. The development of a monoclonal antibody targeting the mutational site H3.3 p.Gly34Trp allowed the development of specific and reliable immunohistochemical diagnostic techniques, acting as a surrogate marker for molecular analysis [31]. Immunoreactivity for the mutated H3F3A is only found in the neoplastic, mononuclear cells, while the giant cells are typically negative. Histological malignancy in GCTB can take many forms. It is usually determined by overgrowth of the mononuclear cells, which also feature increased pleomorphism, spindle morphology and brisk mitotic activity, with findings of atypical mitoses. Malignant GCTB can develop histological characteristics of different high-grade sarcomas, including undifferentiated pleomorphic sarcoma, fibrosarcoma, and osteosarcoma [32]. The term primary malignant GCTB is used for those cases that present nodules of malignant GCTB amidst conventional GCTB in the context of primary disease. Secondary malignant GCTBs are more common and are defined by malignant transformation of a GCTB following treatment [32]. The H3.3A mutation, detected with immunohistochemical or molecular methods, is usually retained in the malignant population [31], but there are cases where it is lost, instead [33]. There is no known specific genetic signature of malignant GCTB. Denosumab treatment is known to cause a spectrum of histological changes in GCTB, including evidence of bone deposition, depletion of the giant cells’ component, and spindling of the mononuclear cells. These changes can be either diffuse to the whole tumour or focal, with persistence of a portion of conventional GCTB [34]. The described changes cause significant overlap with the histological appearance of osteosarcoma or secondary malignant GCTB, making distinction of these entities extremely hard on a histopathological basis.

### 1.3. The Challenge of Imaging: What Change? Radiological Evaluations

The typical radiological features of GCTB include a purely osteolytic lesion, well defined without sclerotic margin, multiloculated, eccentric in location, that extends to the subchondral bone. GCTB may also have aggressive features, such as a wide zone of transition, cortical thinning, expansile remodelling, or even cortical bone destruction and an associated soft-tissue mass.

The GCTB biological behaviour has posed the problem of achieving correct diagnostic and therapeutic management; thus, several staging systems have been proposed over the years. Three grades Campanacci’s classification is based on conventional radiography findings: grade one (latent) relates to a lesion with a well-defined margin, presence of sclerotic border and absence of cortical involvement; in grade two (active), the tumour has well-defined margins but no peripheral sclerosis, with thinned cortical and bone remodelling; in the third grade, the lesion shows indistinct edges, soft tissue infiltration, and erosions of the cortex [35]. Computed tomography (CT) is the method of choice for better delineation of cortical alterations, as it enables multiplanar evaluation and differentiation of solid, necrotic, and cystic components due to the administration of intravenous contrast medium. The magnetic resonance (MR) imaging findings are nonspecific, usually consisting of low/intermediate signal in T1-weighted images and high signal in T2-weighted images that can be variably dishomogeneous due to the fibrous components and the cystic parts. Intravenous gadolinium administration usually shows heterogeneous enhancement of the lesion and permits the differentiation between the cystic and solid components, and MRI is the method of choice to demonstrate tumour extension to the adjacent joint and soft tissue. The radiographic features of primary malignant GCTB are often identical to those of a giant benign cell tumour of bone, and in most cases, it has been impossible to distinguish primary malignant GCTB from a benign lesion on plain films [4,36]. Low Campanacci stage (stage 1) is the only clear distinguishing feature for benign GCTB. CT and MRI have also failed to provide specific signs [5]. Secondary malignant GCTB is difficult to differentiate from recurrent benign GCTB radiologically, but in most cases the presence of Campanacci grade III at the time of diagnosis, with cortical permeation and an associated soft-tissue mass, could reflect its aggressive behaviour [6]. Although the diagnosis of malignant GCTB appears difficult *ab initio*, some radiological changes of the lesion that occur during the follow-up can lead to suspicion of a malignant transformation [37]. Unfortunately, no uniform imaging assessment criteria have been approved to specifically evaluate the response to denosumab treatment in GCTB. The principal signs assessed by MRI or CT, indicative of positive response to denosumab treatment, are usually an increased radiopacity within the area of tumour osteolysis, for the appearance of osteosclerosis, and the construction of marginal neocortex. Considering that, the absence of these peculiar markers and the increase in size of soft-tissue mass during treatment and follow-up, should suggest an aggressive behaviour of benign GCTB, reasons to justify further investigations. The use of 18F-Fluorodeoxyglucose (FDG) positron emission tomography/computed tomography (18F-FDG PET/CT) to assess GCTB response to denosumab treatment is increasing over time. Indeed, the significant reduction in FDG avidity during therapy, which occurs in over 90% of patients, appears to be the most sensitive method for evaluating treatment efficacy and tumour control over time, compared with size/density variation on CT [38,39,40]. In addition, the development of new bone formation can be assessed on the co-registered CT scan. Unlike other benign bone conditions, the peculiarly high FDG avidity of GCTB does not allow the suspicion of malignant transformation merely on the basis of tumour FDG uptake (Figure 1).

However, given the rarity of the lack of response to denosumab in GCTB, persistence or increased uptake of FDG during treatment should lead to a suspicion of malignant transformation, suggesting the need for a new bioptic evaluation (Figure 2).

### 1.4. Malignant GCBT: Current Available Data from Clinical Series

One of the largest retrospective case series of malignant GCTB patients was reported by Bertoni et al. in 2003 [4]. In 17 patients (1.8% of the entire series) a malignant transformation was found within the GCTB. Among them, 5 were primary and 12 secondary malignant GCTB (half of the latter were postradiation sarcomas). Patient age ranged from 20 to 68 years (median, 62 years) for primary, and from 30 to 77 years (median, 40 years) for secondary malignant GCTB. The average latent period between diagnosis of GCTB and diagnosis of secondary malignant GCTB was 9 years (range, 3–15 years) for patients with post-radiation malignant transformation and 19 years (range, 7–28 years) for patients with secondary spontaneous malignant transformation. In both cases, sarcoma was most frequently found in the long bones around the knee joint (three and six cases, respectively), with a preference for the distal femur. The histological examination of high-grade sarcoma in the primary GCTB group showed osteosarcoma in four cases and malignant fibrous histiocytoma in one case. In the secondary malignant GCTB group, the histological examination deposed for osteosarcoma in nine cases, fibrosarcoma in two cases, and malignant fibrous histiocytoma in one case. The outcomes associated with all malignancies in GCTB were poor, with the worst prognosis associated with post-radiation secondary malignant GCTB [4].

A second retrospective analysis was recently published by Liu et al. The authors reported in a retrospective analysis from 1998 to 2016, 1365 patients with extremity GCTB [6]. Thirty-two (2.3%) patients had malignant GCTB, including twelve primary malignant GCTB and twenty secondary malignant GCTB. The distribution of malignant GCTB by anatomical location in this study was similar to that observed by other authors, with the most common sites being the distal femur and proximal tibia, and the most common presenting symptoms being pain and swelling. Radiologically, they presented characteristics as aggressive Campanacci grade III tumours with prominent bone destruction and soft tissue extension. Fifteen out of the twenty cases of secondary malignant GCTB presented histological features of osteosarcoma, four of undifferentiated pleomorphic sarcoma, and one of fibrosarcoma. The mean latent period in patients with secondary malignant GCTB was 7.9 years. Secondary malignant GCTBs were more frequently observed in patients with late local recurrence rather than in patients with early local recurrence, which is usually related to benign GCTB (median: 57 months vs. 19 months), with a reported cut-off time of 4 years. The 5-year survival estimates of primary malignant GCTB and secondary malignant GCTB were 56.2% and 40.0%, respectively (log rank, *p* = 0.188). The risk of local recurrence seems to be associated with adequate margin excision. As resulted in this analysis, local recurrence was more frequent in patients with inadequate margins compared with patients with optimal resection (7 of 9 patients vs. 5 of 24, *p* 0.006). When margins were inadequate, a 12.6 times higher probability of appearance of local recurrence was observed (*p* = 0.008). The incidence of lung metastases was high in malignant GCTB, developed in 22 of 32 patients (69%), of which 19 were metachronous. The median distant metastasis-free survival was 9 and 21 months for malignant and benign GCTB, respectively (*p* = 0.002). After a median follow-up of 2.1 years, they observed, for malignant GCTB, a 5 and 10-year overall survival rate estimate of 45.8% and 36.1%, respectively, that seems to be poorer than the overall survival rate expected in osteosarcoma with current chemotherapy protocols. The challenges in diagnosis of malignant GCTB have significant implications in their surgical management. Chemotherapy was associated with a longer pulmonary metastasis-free survival (13 months vs. 6 months, *p* = 0.002), but not with an increased overall survival (57.0% vs. 33.3%, *p* = 0.167) [33]. With these considerations, an accurate diagnosis is critical to avoid inadequate surgical margins when treating primary malignant GCTB. Furthermore, adjuvant chemotherapy showed the absence of survival benefit but seemed associated with increased pulmonary progression-free survival [6]. The Memorial Sloan Kettering Cancer Center (MSKCC) described the highest rate of primary malignancy with 26 (9.5%) cases among 275 patients after a follow-up of up to 31 years. This difference from other studies in the higher percentage of primary malignancy, can be explained with the application of well-defined diagnostic criteria and the prolonged follow-up [5].

### 1.5. Malignant GCBT and Denosumab

Recently, few analyses reported cases of malignant transformation of GCTB during denosumab treatment (Table 2).

The suspected mechanism of GCTB sarcomatous transformation after denosumab therapy is barely understood, but is probably correlated with its actions against RANKL [41]. Few hypotheses have been proposed to date [42,43,44]. The suspected activity of denosumab on the immune system and on the process of inflammation, can explain the risk of new malignancies as a result of immunosuppression, due to RANKL inhibition, which is involved in lymphocyte development and lymph-node organogenesis. A second point of view concerns the osteosarcoma cells and the effect of RANKL expression in increasing the level of nuclear factor IB (NKIB) [45]. NKIB is an essential transcription factor important for down-regulation of the susceptibility to nuclear oncogenes [46]. In this scenario, the inhibition of RANKL could lead to osteosarcoma carcinogenesis by raising susceptibility to nuclear oncogenes. The last hypothesis suggested that the role of denosumab restraining RANKL is the induction of aberrant osteoblastic differentiation and tumourigenesis through the Sema3A pathway. The Sema3A gene is normally upregulated by RANKL in osteosarcoma, and its deletions could lead to abnormal cartilage and bone growth [46,47,48]. As mentioned above, these questionable hypotheses came from few analyses that reported malignant transformation of GCTB in patients after denosumab treatment. Chawla et al. followed-up in a phase 2 trial, for a median follow-up of 58 months, 526 patients with GCTB that received at least one dose of denosumab and observed five cases with sarcomatous transformation of previously histologically benign GCTB (*n* = 4; 1%) or secondary malignant GCTB (*n* = 1; <1%). In the four patients, the time from diagnosis of GCTB to malignant transformation ranged from 17 months to 11 years. Histologically, these cases presented as undifferentiated spindle cell sarcoma and high-grade osteosarcoma (two patients each). The authors reported that the incidence of confirmed malignant transformation in patients treated with denosumab in their study was similar to that of previous studies in which denosumab was not administered. However, careful and close radiological and clinical evaluation during treatment is warranted, as evidenced by many misdiagnosed patients showing no expected radiological intratumoural calcifications and recurrent or progressive pain, expected as a consequence of the mechanism of action of denosumab in the GCTB lesion. Malignant transformation was more common in patients who had previous radiotherapy than in those who had not received radiotherapy, which, therefore, should be considered during the close monitoring period [8]. Small series reported an incidence of malignant transformation from benign GCTB of 3–4% after denosumab administration, occurring at different times ranging from 8 to 55 months [9,10,11]. The rarity of the cases analysed cannot define the real correlation between denosumab and malignant transformation, and a longer follow-up during denosumab treatment in these patients is needed to understand and confirm the safety of denosumab for GCTB.

## 2. Materials and Methods

We analysed three cases of histologically confirmed malignant transformation of GCTB diagnosed at Regina Elena National Cancer Institute (IRE) in Rome, an Italian referral centre and a EURACAN (European Network for Rare Adult solid Cancer) centre for the treatment of soft tissue and bone sarcomas. Our cases were extracted from a database of 110 patients with GCTB treated in our institution from February 2005 to December 2021. All therapeutic approaches for each patient were discussed with our Sarcoma Multidisciplinary Team (MDT).

### 2.1. Patient 1

In July 2017, for persistent right knee pain, a 29-year-old woman performed clinical and diagnostic exams, including an MRI that showed the presence of an osteolytic lesion of the distal femur with sharp, sclerotic margins, without apparent interruption of the cortical bone and/or involvement of the perischeletric soft tissues (Figure 3A,B). In August, the patient underwent bioptic procedure of the osteolytic lesion at the right distal femur. On microscopic examination, the lesion (Figure 4A) showed a proliferation of osteoclast-like giant cells admixed with mononuclear cells. A pathological diagnosis of GCTB was formulated. After discussion with our MDT, the patient underwent intralesional surgery with curettage of the lesion and bone grafting. Subsequent follow-up was negative until January 2018, when a 18F-FDG-PET showed focal hypermetabolism compatible with relapse of the disease in the femoral shaft (standardised uptake value (SUV) max 11.1; Figure 1) and an involvement of soft tissues at the distal third of the right thigh. Furthermore, MRI confirmed the presence of an oval lesion (6.9 × 4.7 × 5.9 cm) at the antero-medial level of the distal right femur, compatible with local regional relapse (Figure 3C,D). A biopsy of the suspect lesion was performed, and histological analysis confirmed GCTB diagnosis, including a higher degree of atypia of the neoplastic cells and depletion of the giant cells. For these reasons, the patient started neoadjuvant treatment with denosumab with close clinical and radiological monitoring. The treatment with denosumab was continued for about 4 months until May 2018, when a new CT scan showed a significant increase in soft tissue in the perischeletrical area with calcified concamerations. Therefore, the patient underwent an extraarticular resection of the right knee and subsequent reconstruction with a silver modular oncological prothesis. Histological analysis showed unequivocal features of malignancy, with the neoplasm consisting of markedly atypical spindle cells diffusely infiltrating the native bone. A diagnosis of secondary malignant GCTB with transformation into fibroblastic osteosarcoma was initially formulated. Consequently, since November 2018, the patient was treated with adjuvant chemotherapy including high-dose methotrexate, adriamycin, and cisplatin (MAP regimen) for 27 weeks plus mifamurtide [49]. The follow up was negative until November 2020 when, after continuous pain in the proximal right humerus, a 18F-FDG-PET showed a pathological alteration of the right humerus (SUV max 7.2) and a subsequent MRI showed a proliferative lesion of the osteogenic mesenchymal series at the level of the right humerus. Considering the presence of only one suspicious lesion, a histological confirmation with biopsy was needed. The histological examination of the lesion (Figure 4C) showed a crowded proliferation of atypical, round cells in the context of a richly vascularised stroma, with no giant cells. Meanwhile, immunostaining for the H3-3A mutation was performed, giving negative results (Figure 4D) [31]. The neoplastic lesions were instead immunoreactive for SATB2, a marker of osteoblastic differentiation, and a final diagnosis of osteosarcoma was, therefore, formulated. After this, the patient was treated with ifosfamide as first line, with gemcitabine-docetaxel as second line and subsequently with pazopanib, until her death in January 2022 [50,51,52,53]. The OS was 53 months from first GCTB diagnosis, 43 months from malignant histopathological transformation, and with a time to malignant transformation of 9 months. In the context of this study, a retrospective review of the histological slides for all the subsequent bioptic and surgical samples was performed, along with the immunohistochemistry for H3F3A. All lesions, including the older one (Figure 3B) tested negative for H3F3A and positive for SATB2, a finding suggesting molecular features of giant-cell-rich osteosarcoma from the beginning, despite the morphological pattern apparently suggestive of benign GCTB.

### 2.2. Patient 2

In October 2016, a 48-year-old man underwent clinical and radiological investigations because of worsening pain at his right knee. The RXs of the right knee showed a large radiolucent lesion involving the tibial plateau and the proximal third of the tibial shaft, for a total of 10 cm, with thinning of the cortical bone (Figure 5A,B). For this reason, in November 2016, the patient underwent a biopsy of the lump on the right proximal tibia. Histopathological examination of this lesion (Figure 6A) showed a proliferation of osteoclast-like giant cells admixed with mononuclear cells. The two populations showed mild nuclear atypia and a diagnosis of GCTB was formulated. Despite the huge dimensions of the lesion, both in terms of size and extension to the cortical bone, the patient underwent curettage of the tumour, local adjuvant and bone grafting. Ever since, the patient started regular, negative clinical and radiological checks until July 2020 when a US of the right knee showed a voluminous inhomogeneous area of doubtful nature, suspected of local relapse (maximum diameter 2.5 cm). The following MRI of the right proximal tibia and 18F-FDG-PET (Figure 2) confirmed the presence of pathological tissue with high metabolic activity at the proximal right tibia suggestive for local recurrence (SUV max 26.5). In October 2020, a biopsy of the lesion of the proximal right tibia confirmed the hypothesis of recurrence of GCTB. Immunoreactivity for mutant H3F3A and SATB2 supported the diagnosis. Retrospective immunostaining for H3F3A in the 2016 biopsy was also performed and tested positive (Figure 5B). After discussion in our MDT, the patient received neoadjuvant treatment with denosumab for 3 months [35]. At the clinical and radiological evaluation, in January 2021, the RX of the right leg showed the presence of a lytic alteration of the proximal meta-epiphyseal region with interruption of the cortical bone on the medial aspect of the right tibia, compatible with loco-regional relapse (Figure 5C,D). The hypothesis of recurrence was confirmed by an MRI of the right leg that observed a portion of pathological tissue in the meta-epiphyseal region of the tibia with extraskeletal involvement (8.1 × 10 × 7.7 cm) (Figure 5E,F). A 18F-FDG-PET confirmed the progression of the known lesion of the right leg (SUV max 21.4) without distant metastases. In February 2021, the patient underwent extraarticular resection of the knee and subsequent reconstruction with oncological megaprosthesis; the extensor apparatus was restored by rotating the medial twin muscle. Histological examination of the new lesion (Figure 6C) showed proliferation of atypical spindle cells in the context of fibrous stroma, with no giant cells. The neoplastic cells were immunoreactive for H3F3A (Figure 6D) and SATB2. The histopathological picture was equivocal, possibly representing either denosumab-associated changes or the development of secondary malignant GCTB in the recurrent lesion. In light of the clinical and radiological characteristics, the latter hypothesis was deemed more likely. Therefore, from April to December 2021 the patient received adjuvant chemotherapy including adriamycin, cisplatin, and ifosfamide, according to the EUROBOSS scheme [54]. At the last radiologic evaluation, the 18FDG PET scan was negative for relapse. Currently, the patient continues periodic follow-up checks with clinical and radiological evaluations, with an OS of 64 months from GCTB diagnosis, 13 months from malignant transformation, and a time to malignant transformation of 51 months.

### 2.3. Patient 3

In July 2010, a 20-year-old woman, with a history of worsening pain in the left hip, was referred to the oncological orthopaedic division of our Institute. Plain X-ray and subsequent CT SCAN, revealed an expansive osteolytic lesion in the proximal left femur highly suggestive of GCTB (Figure 7A,B). In September 2010, the patient underwent biopsy of the lesion. At microscopic examination, diagnosis of GCTB was formulated (Figure 8A). A curettage of the lesion and subsequent bone grafting was performed, and the histological specimen confirmed the previous GCTB diagnosis. The subsequent follow-up was negative until April 2011 when, for the occurrence of a pathological femoral fracture, the patient performed a chest and lower skeletal segment CT SCAN that showed a large area of osteostructural alteration of the neck and femoral shaft, with pathological fracture and newly formed tissue that completely subverted the neck and pretrochanteric region (Figure 7C–E). A subsequent 18F-FDG-PET confirmed the presence of relapse with high glycolytic metabolism at the level of the left proximal femur with extension to the pretrochanteric soft tissues (SUV 18.4). In June 2011, in suspicion of an aggressive behaviour, the patient underwent biopsy of the lesion, with evidence of proliferation of epithelioid cells with moderate-to-severe atypia, large nuclei and clear or dusty cytoplasm. No giant cells were present. The histopathological examination was suggestive of a secondary malignant GCTB in the recurrent lesion, with features of osteosarcoma. An increase in serum beta-human chorionic gonadotropin (βhCG) levels was concomitantly documented. Once excluding a pregnancy with no gestational sac seen at transvaginal ultrasound, and considering that cases of βhCG raise in presence of osteosarcoma are reported in literature, we concluded for a paraneoplastic ectopic secretion [55,56,57,58]. In June 2011, the patient started neoadjuvant chemotherapy including high-dose methotrexate, adriamycin, and cisplatin (MAP regimen) and a significant decrease in serum level of βhCG was observed [54,59]. Unfortunately, with rapidly metastatic progressive disease, the patient passed away in November 2011, with an OS of 14 months from GCTB diagnosis, of only 5 months from malignant transformation, and with a time to malignant transformation of 9 months [60]. In the context of this study, immunostains for H3F3A and SATB2 have been retrospectively performed on all histological samples from this patient, and always tested positive, confirming all previous formulated diagnoses (Figure 8B,D).

## 3. Discussion

Early diagnosis of malignant transformation of benign GCTB can be challenging because of the rarity of this occurrence (less than 1% of patients with GCTB) and also because of the high risk of pathological misdiagnosis, significantly reduced by the relatively recent introduction of H3F3A immunohistochemistry, typically positive in benign GCTBs and which may or may not be lost following malignant transformation [31,33]. Furthermore, primary malignant GCTB can be characterised by focal areas of atypia in the context of a tumour that is otherwise composed of mostly typical cells, meaning that a correct diagnosis can only be formulated if the biopsy is performed in one of the former areas. Consequently, a late diagnosis of malignant GCTB is a very significant risk and can significantly impact the patient’s prognosis. Historically, malignancies in GCTB have been observed after radiotherapy, but previous data have shown how they may also occur after surgical treatment, such as bone grafts, without adjuvant radiotherapy. Another aspect that deserves to be analysed but is difficult to assess, considering the few data available in the literature, is the association between GCTB malignant transformation and primary tumour location. In our monocentric experience, it does not seem to be correlated, but more data are needed to highlight this point. In this scenario, a clearer understanding of the incidence and behaviour for which benign cells of GCTB turn into malignant cells is crucial to manage and follow these patients. In our reported cases, all patients presented local recurrence of their disease after surgical treatment, with the new lesions presenting signs of histological dedifferentiation. All patients presented a shorter time to malignant transformation, as usually reported by the literature (Table 3). This can partly be explained by the prompt needle CT-guided biopsy performed on patients immediately after detection of a suspicious lesion on imaging. We also must keep in mind that the possibility of pathological misdiagnosis, as seen in the first case, can be the underlying cause for different times to relapse in different case series, with both malignant and benign histological mimics having completely different clinical histories. We reported in our cases the first young patient with malignant transformation of GCTB after recurrence and 4 months of neoadjuvant treatment with denosumab. As observed before, there have been some concerns about treatment with denosumab and the possible risk of GCTB malignant transformation, but an early detection of this complication with accurate imaging and needle TC-guided biopsy could bypass a potential misdiagnosis. In the phase II trial reported by Chawla et al., denosumab showed clinical benefit in patients with GCTB, and few cases with primary malignancies; it was a sub-analysis of a phase II study [7,8], whose primary endpoint was to evaluate the safety of denosumab treatment. To be more specific, between adverse events emerging during treatment, malignant transformation of GCTB was an event of interest. In fact, 20 (4%) of 526 patients with a potential diagnosis of malignancy were identified, but, after pathologic reanalysis of original biopsy material, 15 of 20 were confirmed to be misdiagnosed from the start, while only five patients (four and one) were found to be cases of true malignant transformation: four (1%) cases from histologically benign GCTB and one (<1%) case of secondary malignant GCTB [7]. Even Rutkowski et al. reported in a phase II study the clinical benefits of denosumab in 222 patients with GCTB. In this analysis, two cases developed malignant transformations associated with previous radiotherapy several times, and two other cases progressed during long exposure of denosumab treatment. Perhaps, as suggested by the investigators, the initial biopsy was misdiagnosed at the beginning and was likely correlated with primary malignant GCTB. Anyway, in these trials, confirmed diagnoses by histology, or morphology about the GCTB sarcoma transformation correlating to denosumab treatment, do not exist. Unfortunately for our patient, sarcoma diagnosis was confirmed 9 months after the primary diagnosis of GCTB, and the prognosis was inauspicious, with the appearance of metastases during adjuvant treatment and quick progression subsequent to several chemotherapy lines [3]. The second patient, currently in follow-up, is showing a favourable outcome even though the malignant transformation was diagnosed more than 4 years after the primary GCTB. In this case, after curettage and first relapse for GCTB, the patient was treated with neoadjuvant denosumab for only 3 months and at that point the histological examination following surgery revealed malignant transformation. Considering that imaging showed the presence of pathological tissue with high metabolic activity before starting denosumab, the transformation may have already been present before the beginning of treatment, with the biopsy failing to sample the malignant portion of the tumour. Despite the evolution of disease, the patient finished the adjuvant treatment as per EUROBOSS protocol and is now in follow-up with an absence of recurrence or metastases. The last case involved a young woman with diagnosis of GCTB and subsequent transformation in high grade osteosarcoma several months after curettage. Interestingly, the patient showed an increased level of βhCG during follow-up, and as other pregnancy exams were negative, a paraneoplastic etiology was suspected and eventually confirmed by decreasing values during chemotherapy. Few cases have been reported about the correlation of ectopic βhCG and sarcomas [55]. Masrouha et al. reported a series of 32 tumours analysed, in which five were found to be positive for βhCG expression (one strongly and four weakly). In these cases, the incidence of βhCG expression was correlated with a poor prognosis and weak response [56]. The real role of βhCG expression in carcinogenesis is still unknown, but in several analyses it resulted associated with worse response to chemotherapy and poor prognosis, as per our patient.

## 4. Conclusions

With the variants in the clinical characteristics, treatment, and oncologic outcomes observed in these cases and in retrospective analyses, the behaviour of malignant GCTB is unclear and unpredictable, and it is difficult to draw conclusions to guide treatment and subsequent follow-up. An early detection with accurate imaging and needle CT-guided biopsy in the case of suspicious lesions should be considered to avoid misdiagnosis.

## Figures and Tables

**Figure 1 ijms-23-10721-f001:**
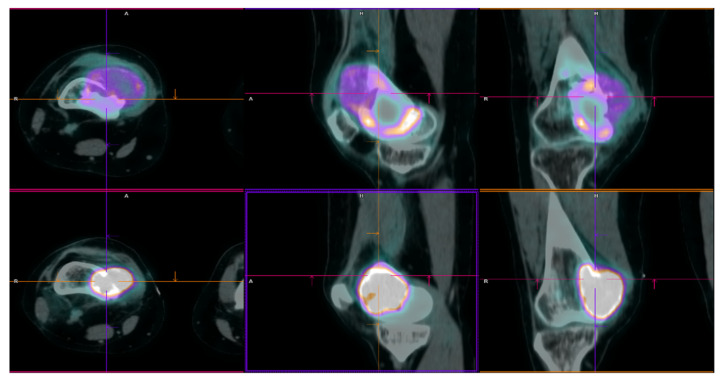
^18^F-FDG PET/CT axial (**left**), sagittal (**middle**) and coronal (**right**) fused view of right knee in patient no. 1. The first PET/CT performed at time of first relapse of GCTB (lower row) showed a focal area of intense FDG uptake (SUVmax = 21) corresponding to a lytic lesion of the medial meta-epiphyseal aspect of the right femur, showing extension to the neighbouring soft tissue. A second PET/CT scan (upper row), showed a disease relapse with an FDG avidity lower than original lesion (SUVmax = 11). A = anterior; R = right; H = high.

**Figure 2 ijms-23-10721-f002:**
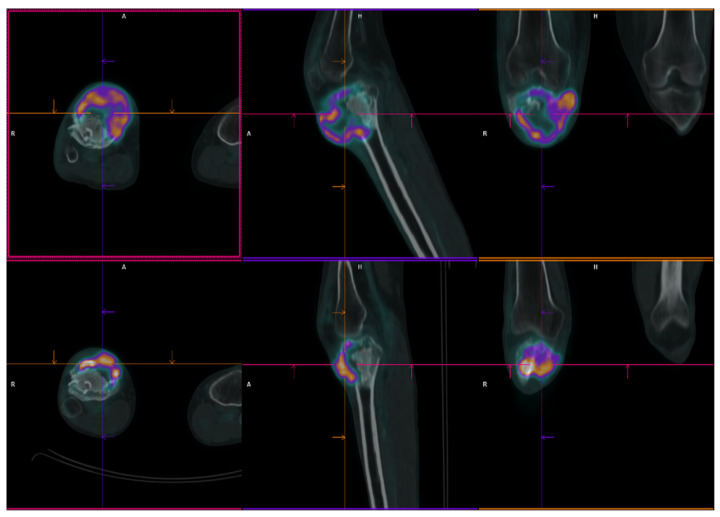
^18^F-FDG PET/CT axial (**left**), sagittal (**middle**) and coronal (**right**) fused view of right knee in patient no. 2. A first PET/CT scan (lower row) showed the presence of a pathological tissue with high metabolic activity (SUVmax = 26.5) at the level of the right tibia extending to the neighbouring soft-tissue compatible with relapse. A second PET/CT scan performed three months after starting treatment with denosumab (upper row), showed a significant increase in soft-tissue component of the lesion, with stability of FDG uptake. A = anterior; R = right; H = high.

**Figure 3 ijms-23-10721-f003:**
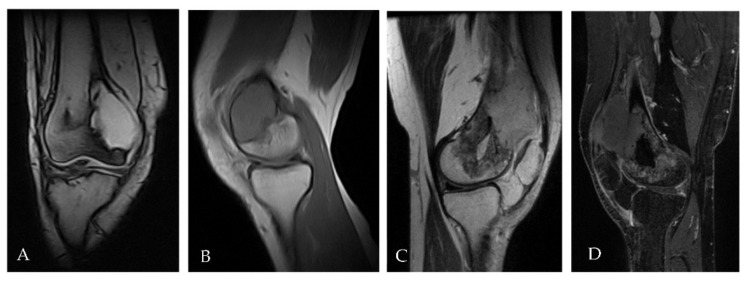
(**A**,**B**) MRI performed at time of initial diagnosis showed a focal eccentric lesion located in the meta-epiphyseal of distal right femur without interruption of cortical bone and/or involvement of neighbouring soft tissue. (**C**,**D**) MRI performed after surgery with curettage six months later showed the presence of solid heterogeneous high signal tissue in T2 (3C) enhancing post contrast (3D) with large extension in the soft tissue compatible with relapse.

**Figure 4 ijms-23-10721-f004:**
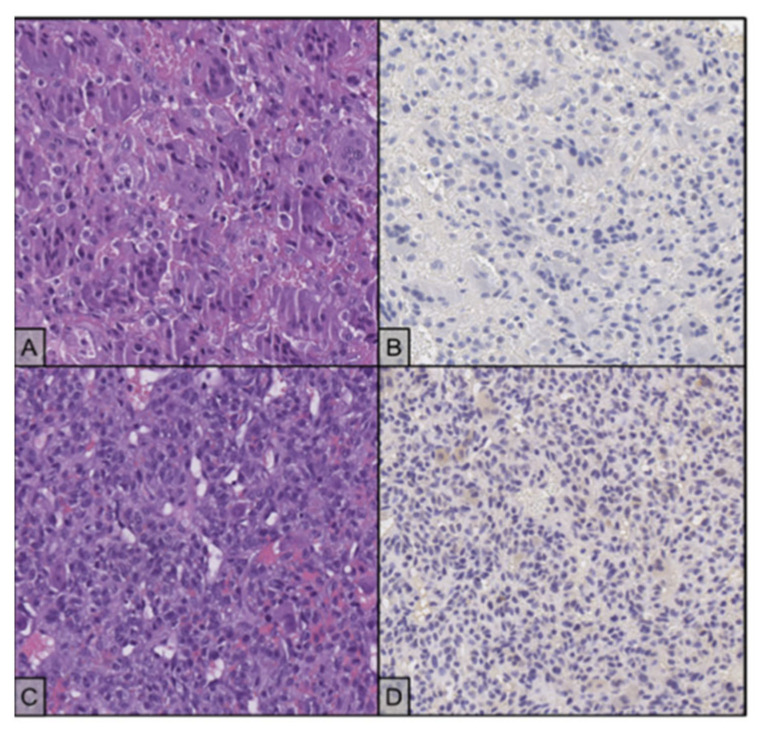
(**A**) The lesion is composed by multinucleated and mononuclear cells (H&E stain, 20×); the giant cells have hyperchromatic, homogenous nuclei, while the mononuclear cells tend to present slightly larger nuclei with finely dispersed chromatin and evident nucleolus. (**B**) Both cell populations test negative for H3F3A immunostaining (20×). (**C**) Recurrent disease shows proliferation of atypical, round cells, with no evidence of giant cells. (H&E stain, 20×). (**D**) H3F3A immunostaining is negative in the neoplastic cells (20×).

**Figure 5 ijms-23-10721-f005:**
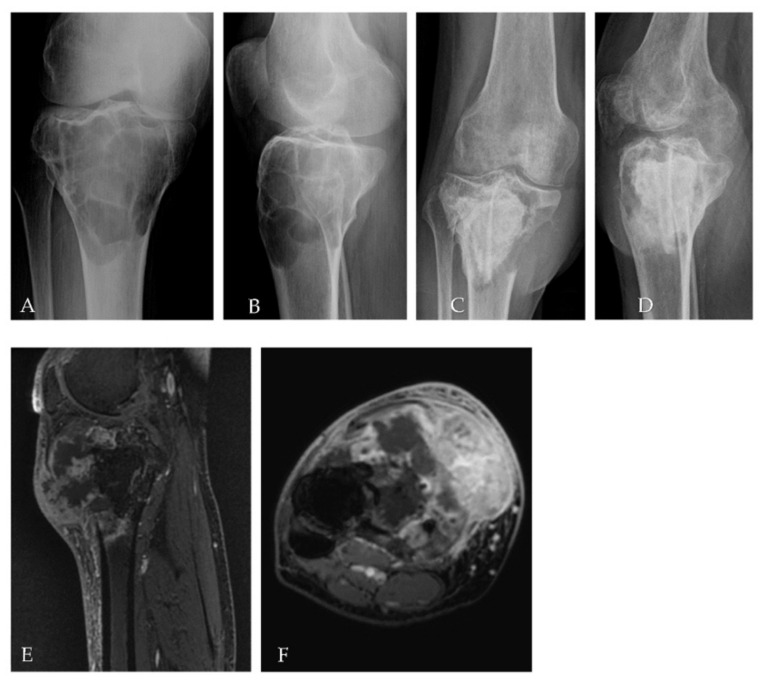
(**A**,**B**) The RXs of the right knee performed at time of initial diagnosis, showed a large, well-defined osteolytic lesion involving the tibial plateau and the proximal third of the tibial shaft, with thinning of the cortical bone. The RX (**C**,**D**) of the right leg performed after neoadjuvant treatment with denosumab showed the presence of a lytic alteration of the proximal meta-epiphyseal region with interruption of the cortical bone on the medial aspect of the right tibia; an MRI (**E**,**F**) confirmed the present of solid inhomogeneous tissue in the meta-epiphyseal of the tibia with significant involvement of extraskeletal soft tissue.

**Figure 6 ijms-23-10721-f006:**
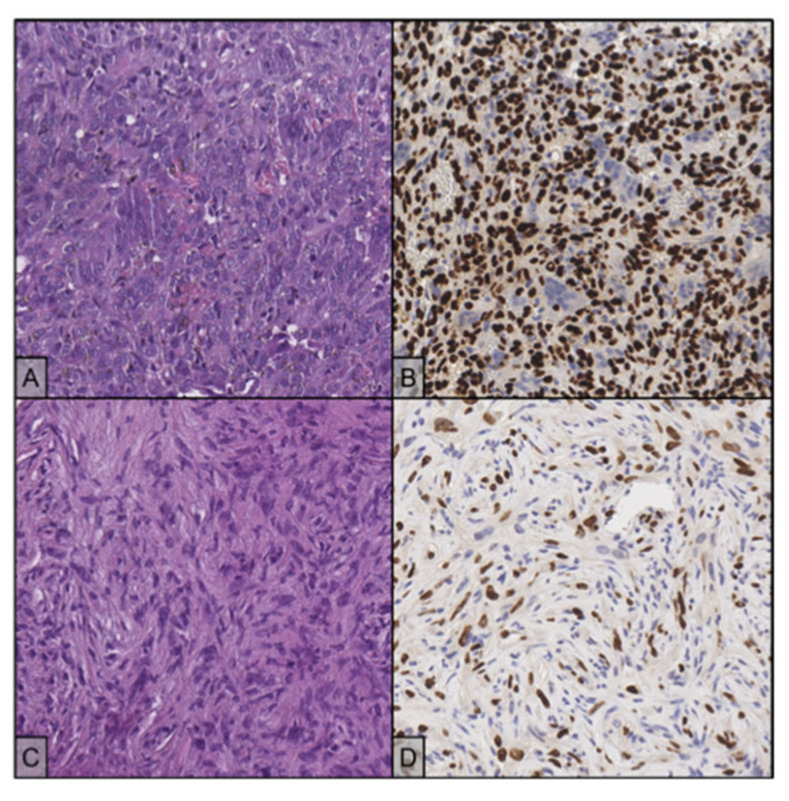
(**A**) Typical histology with osteoclast-like giant cells admixed with mononuclear cells (H&E stain, 20×). (**B**) Mononuclear cells show immunoreactivity for H3F3A, while the giant cells are negative (20×). (**C**) Recurrent disease shows different morphological features from the primary neoplasm, with proliferation of an atypical spindle cell population and no evidence of giant cells. (H&E stain, 20×). (**D**) Neoplastic cells maintain focal positivity for H3F3A immunostaining (20×).

**Figure 7 ijms-23-10721-f007:**
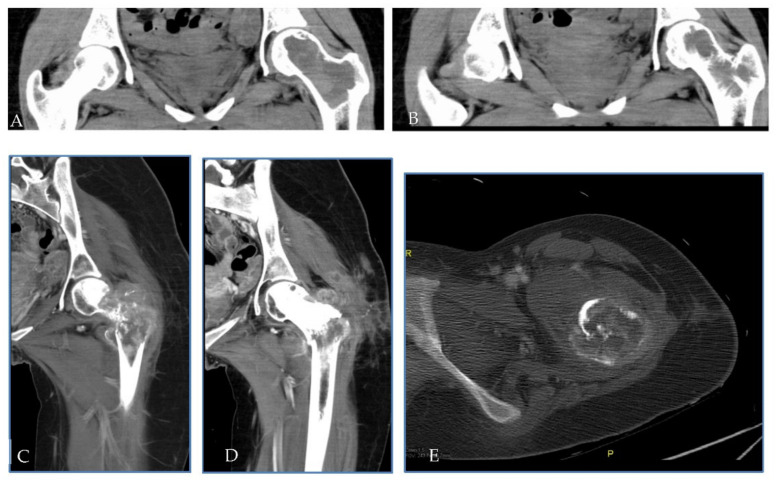
(**A**,**B**) At time of diagnosis, CT scan showed a well-defined osteolytic lesion of the head and neck of the left femur with integrity of cortical bone. (**C**–**E**) During the follow up, for the occurrence of a pathological fracture, CT scan showed a large, complex area of osteostructural alteration of the neck and femoral shaft composed of solid vascularised tissue widely extended in perifemural tissue.

**Figure 8 ijms-23-10721-f008:**
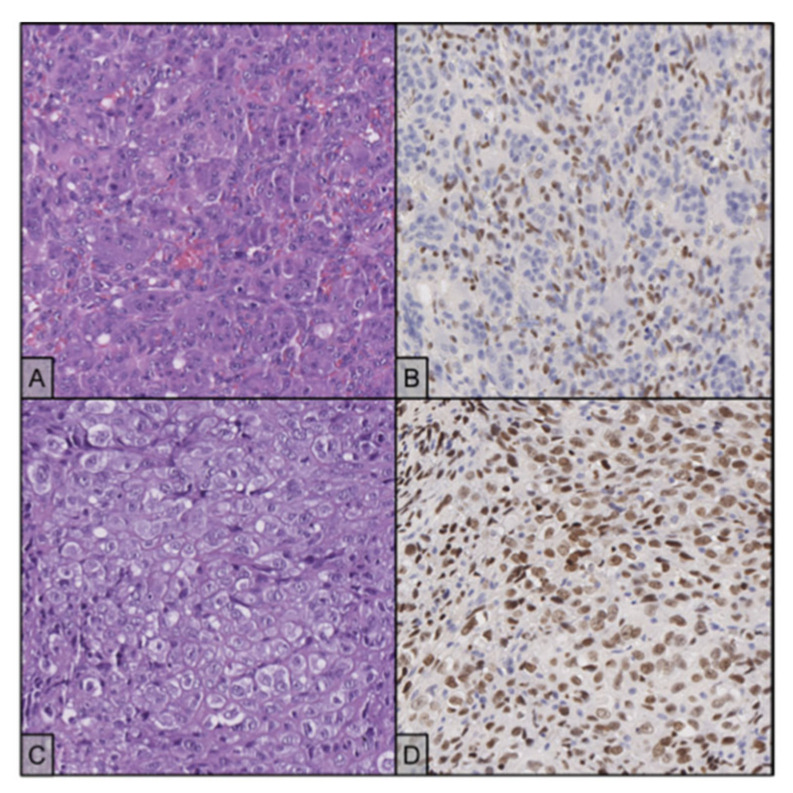
(**A**) The primary disease shows typical histology with osteoclast-like giant cells between numerous mononuclear cells (H&E stain, 20×); the two populations showed remarkably similar nuclear features, delicate nucleoli and finely dispersed chromatin; the interposed stroma appeared haemorrhagic. (**B**) Mononuclear cells react with an antibody against H3F3A, while the giant cells are negative (20×). (**C**) Recurrent disease shows different morphological features and no evidence of giant cells (H&E stain, 20×). (**D**) Neoplastic cells maintain positivity for H3F3A immunostaining (20×).

**Table 1 ijms-23-10721-t001:** Summary of published studies reporting malignant transformation after histological diagnosis of GCTB.

Ref.	NoP	Malignancies *n* (%)	Primary	Secondary	Time to Malignant Transformation	FU
Rutkowski et al. [3]	222	4 (1.8%)	-	4	3.5 yr	Median 13 mo
Bertoni F. et al. [4]	924	17 (1.8%)	5	12	Average l9 yr	NR
Domovitov SV et al. [5]	275	31 (11.3%)	26	5	NR	31 yr
Liu W et al. [6]	1365	32 (2.3%)	12	20	Secondary 7.9 yr Primary: NR	9.5 yr
Chawla et al. [7,8]	526	5 (1%)	4	1	74.5 mo	Median 58.1 mo
Agarwal et al. [9]	25	1 (4%)	-	1	8 mo	27 mo
Treffel et al. [10]	35	1 (2.9%)	-	1	18 mo	NR
Perrin et al. [11]	25	1 (4%)	-	1	55 mo	Median 57 mo
Palmerini et al. [12]	532	14 (2.6%)	5	9	7.8 yr	58 mo

FU: follow-up; GCTB: giant cell tumour of bone; mo: months; n: number; NoP: numbers of patients; NR: not reported; Ref: references; yr: years.

**Table 2 ijms-23-10721-t002:** Extracted data from patients reporting malignant transformation following denosumab.

Ref.	NoP	Transformation *n* (%)	Time to Malignant Transformation From GCTB	FU
Rutkowski et al. [3]	222	2 (0.9%)	8.6 mo	Median 13.0 mo
Chawla et al. [7,8]	526	5 (<1%)	Range, 17 mo–11 yr	Median 58.1 mo
Agarwal et al. [9]	25	1 (4%)	8 mo	27 mo
Treffel et al. [10]	35	1 (2.9%)	18 mo	NR
Perrin et al. [11]	25	1 (4%)	55 mo	Median 57 mo
Palmerini et al. [12]	526	14 (2.6%)	7.8 yr	58 mo

All patients received 120 mg of denosumab q28, plus loading dose with 120 mg on D8 and D15 FU: follow-up; GCTB: giant cell tumour of bone; mo: months; n: number; NoP: numbers of patients; NR: not reported; Ref: references; yr: years.

**Table 3 ijms-23-10721-t003:** Summary of clinical cases.

	Age	Sex	ECOG PS	Diagnosis (Date)	Site of Primary GCTB	Surgery	Rec. Date	Treat. Management	Malignant Transformation (Date/Surgery or Biopsy)	Treat.	Evolution	Subsequent CHT Lines	FU/D	Time to MT (Months)	OS from GCTB Diagnosis (Months)	OS from MT (Months)
Pt 1	29	F	0	17 July	Right femur	Curettage and bone grafts	18 February	Neoadj denosumab (7 cycles)	18 July Extraarticular resection: high-grade fibroblastic osteosarcoma	ISG/OS-2	PD	I: IFO II: GEM.TXT III: PAZO	D 22 January	9	53	43
Pt 2	48	M	0	16 November	Right tibia	Curettage and bone grafts	20 October	Neoadj denosumab (6 cycles)	21 February Extraarticular resection: malignant transformation of GCTB	EURO BOSS	NED	-	FU 21 December	51	64	13
Pt 3	20	F	0	10 September	Left femur	Resection of the femoral head, curettage and bone grafts	11 March	Biopsy	11 June Biopsy: high-grade osteosarcoma G3 with aberrant expression of beta-HCG	ISG/OS-1	PD	-	D 11 November	9	14	5

GCTB: giant cell tumour of bone; Pt: patient; PS: performance status; CHT: chemotherapy; Rec. date: recurrence date; Treat: treatment; Treat. Management: treatment management; FU: follow-up; D: death; OS: overall survival; F: female; M: male; ISG/OS-2: Italian Sarcoma Group/Osteosarcoma-2 [49]; IFO: ifosfamide [50]; GEM-TXT: gemcitabine-taxotere [51]; PAZO: pazopanib [52,53]; EUROBOSS: EUROpean Bone Over 40 Sarcoma Study [54]; ISG/OS-1: Italian Sarcoma Group/Osteosarcoma-1 [59]; PD: progression disease; NED: no evidence of disease; neoadj: neoadjuvant; MT: malignant transformation.

## Data Availability

The review did not report conclusive data; therefore, there are no details regarding where data supporting the reported results can be found.

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
