# Peer review of "Malignant Transformation of Giant Cell Tumour of Bone: A Review of Literature and the Experience of a Referral Centre"

_ijms, 2022, doi:10.3390/ijms231810721_

Round 1
Reviewer 1 Report
Interesting paper in a hot topic in musculoskeletal oncology.
I suggest some improvements:
- A key-point is that a follow-up is needed, especially during denosumab treatments. I suggest You to deepen and specify this concer. Which imaging follow-up is suggested? (only PET-CT with an increased SUV? consider other modalities!); I suggest You to read and consider this paper at this regard PMID: 32335707.
An increased dimension of the tumor at follow-up (MRI or CT assessed) together with the absence of the expected denosumab(therapy)-related changes (e.g. intratumoral mineralization) should be taken into account as a possible sign of transformation. Cortical desruption and soft-tissue involvement should be considered as well.
- Please, enrich the iconographic section of the paper. You reported only PET-CT. Since only 3 cases are presented, show their appearance on available cross-sectional (MRI / CT) or plain radiology modalities, with particular regards to the changes related to malignant trasformation.
Author Response
Dear reviewer,
Many thanks for your interesting comments and suggestions that improved the manuscript.
A new comments have been made in the revised manuscript about radiological assessments during the follow-up, with specific attention to imaging changes during denosumab treatment and for suspicious sign of malignancies.
Iconographic section have been enriched as weel. Specifically, radiological imaging for patient 1, 2 and 3 have been added, highlighting the signs of differences from the benign diagnosis to malignant evolution of the lesion.
English language has been checked again and few corrections have been made.
Please see in the attachment the new version of the manuscript with all new comment and corrections marked for your revision.
Kind regards.

Reviewer 2 Report
Dear editor,
Thanks for giving this opportunity to review the manuscript entitled “
Malignant transformation of giant cell tumor of bone. A review of literature and the experience of a referral centre.” by Sabrina Vari , et al. It is generally accepted that malignant giant cell tumor of bone (GCTB) is rare, and accounts for 4% of of all GCTB. Primary malignant GCTB comprises sarcomatous growth juxtaposed to benign GCTB, whereas secondary malignant GCTB comprises sarcomatous growth at the site of previously documented benign GCTB. It presents a big challenge to diagnose primary malignancy, as it may be detected only retrospectively when specimens were re-evaluated. Secondary malignancy risk is believed to be associated with specific previous GCTB treatments, such as radiation, surgery or the use of denosumab. In fact, the history of this malignant transformation is unclear and unpredictabl. In this study, the author presents three cases of malignant transformation of GCTB and reviewed the published data of GCTB malignant transformation, and recommended a strict follow up to detect early malignant transformation. Specific comments regarding the manuscript are as below:
1. From the published data, is tumor location related to GCTB malignant trasformation? That is to say whether there is malignancy incidence difference between the extremity and the axial skeleton.
2. Likely, the question remains whether malignant transformation is correlated with Campanacci’s classification? Is grade III tumor more prone to turn into malignancy than grade I?
Taken together, this is a well-writtend manuscript.
Author Response
Dear reviewr,
thanks for your interesting comments.
To repley to the first comment, unfortunatelly, is not easy to evaluate if exist an association between GCTB malignant transformation and primary benign tumour location, considering few data available in literature. In our experience does not seems to be a correlation, but can be fairly suggested for a new multicentric retrospective analysis.
Likely, for comment 2, there are no data in literature that confirm a correlation with malignant transformation and Campanacci’s classification. But, as described in the manuscript, the presence of Campanacci grade III at the time of diagnosis, with cortical permeation and an associated soft-tissue infiltration, could reflect an aggressive behaviour and possible evolution in malignant tumour, rather than grade I or II.
New comments and English check have been made. Please see the attachment with the new version of the manuscript with all corrections and new comments marked.
Kind regards.

Round 2
Reviewer 1 Report
I am satisfied with the revisions performed
Thank You